

# A Global Multi-Source Tropical Cyclone Precipitation (MSTCP) Dataset

Gabriel Morin[1], Mathieu Boudreault[1], and Jorge L. García-Franco[2, 3]

[1]Department of Mathematics, Université du Québec à Montréal, Canada
[2]Lamont-Doherty Earth Observatory, Columbia University, USA
[3]Escuela Nacional de Ciencias de la Tierra, Universidad Nacional Autónoma de México, México

**Correspondence:** Jorge L. García-Franco (jorge.garciafranco@encit.unam.mx)

**Abstract.** Tropical cyclone precipitation (TCP) has significant impacts on coastal communities through its modulation of flood event frequency as well as the water supply in many regions of the world. Satellite estimates provide our most reliable observations of TCP available globally, however, satellite precipitation estimates were limited because most products only have coverage starting in 1997. This paper presents a dataset of global TCP derived from two publicly available datasets. First, global precipitation estimates were taken from the newly developed high-resolution Multi-Source Weighted-Ensemble Precipitation, version 2 (MSWEP V2) which spans from January 1979 to date. TC tracks were obtained from the International Best Track Archive for Climate Stewardship (IBTrACS) dataset. This Global Multi-Source Tropical Cyclone Precipitation (MSTCP) dataset is comprised of two main files in the format of tables: the main and profile datasets, both from January 1979 to February 2023. The main file provides various TCP statistics per TC track, including mean and maximum precipitation rates. The profile dataset comprises the azimuthally averaged precipitation every 10-km away from the center of each storm (until 500 km). The case study of Hurricane Harvey is used to show that MSWEP estimates agree well with another commonly used satellite product. The main statistics of the dataset are analyzed as well, including the differences in the dataset metrics for each of the six TC basins and for each Saffir-Simpson category for storm intensity. The dataset es freely available at: https://zenodo.org/records/10105751 with the doi doi:10.5281/zenodo.10105751.

## 1 Introduction

Heavy rainfall from landfalling tropical cyclones (TCs) causes devastating impacts such as loss of life, damages on property, infrastructure and supply chains (Villarini et al., 2014; Needham et al., 2015; Lenzen et al., 2019). The impacts from flooding observed during the landfall of Hurricane Harvey in 2017 (Risser and Wehner, 2017; Smiley, 2020) and the remnants of Hurricane Ida in 2022 (LeComte, 2022; Dong et al., 2022) are recent examples of the destructive potential of tropical cyclone precipitation (TCP). These flood-related impacts from TCP are exacerbated by local vulnerabilities (Dominguez et al., 2021) and by the occurrence of other extreme weather events, also known as compound events (Wu et al., 2022).

TCP is not only a hazard but also an important contributor to total rainfall, and therefore water supply, since TCs can account for more than 35% of the annual rainfall over Australia, Taiwan, the Philippines and Mexico (Breña-Naranjo et al., 2015; Khouakhi et al., 2017). For this reason, inter-annual and seasonal variability of TCP can lead to variability in drought




indices (Misra and Bastola, 2016; Dominguez and Magaña, 2018). This means reliable observations of TCP are therefore paramount to understand the role of TCs in local and global water budgets, and the relationship between TCP, floods and societal impacts.

Observations of precipitation have allowed studies to gain a better understanding of the physical processes that control TCP, as well as their impacts. Observational estimates of TCP are essential for studies seeking to understand its environmental

controls (Lin et al., 2015), its spatial distribution and contribution to the total annual rainfall and extreme precipitation (Prat and Nelson, 2016), trends in TCP (Touma et al., 2019; Tu et al., 2021), the response to radiative forcing (Emanuel and Sobel, 2013), as well as model evaluation (Kim et al., 2018b; Vannière et al., 2020; García-Franco et al., 2023). The geographical coverage and focus of research studies also varies as some studies focus on regional TCP (Breña-Naranjo et al., 2015; Touma et al., 2019) whereas other studies have conducted global analyses (Skok et al., 2013; Khouakhi et al., 2017).

TCP can be estimated from observational estimates of precipitation such as gridded-station data, aircraft in-situ data, re-analysis or satellite retrievals. Gridded-station data provide observations of TCP that can extend as far back as the early 1900s (Touma et al., 2019; Niu et al., 2022). However, high-density station data with long coverage can only be found in a few countries and regions. Aircraft radar data is even more limited as only a small number of cases have been sampled in recent decades (Didlake and Houze, 2013). Reanalysis data has reasonable spatial and temporal coverage but precipitation is simulated by the

driving forecast model. Jones et al. (2021) found that reanalyses have a higher discrepancy in their representation of TCP than in total precipitation, which is likely due to their different assimilation techniques and the fact that precipitation is computed through model parametrizations.

Satellite precipitation retrievals are therefore the best option for continuous and reliable monitoring and research of global TCP (Cheung et al., 2018). For instance, the Tropical Rainfall Measurement Mission (TRMM) Multisatellite Precipitation

Analysis (TMPA) dataset, launched in 1997, has been extensively used as benchmark of total precipitation in the tropics and TCP due to its relatively high resolution (0.25°) and temporal (3-h) resolution (Huffman et al., 2007). TCP was estimated in TRMM by dozens of studies using similar methodologies (e.g. Skok et al., 2013; Prat and Nelson, 2016; Khouakhi et al., 2017). However, the satellite was discontinued in 2015 and the dataset ends in 2019 (Huffman, 2016).

The Multi-Source Weighted-Ensemble Precipitation, version 2 (MSWEP V2) is a recent global gridded precipitation dataset

spanning 1979–present (Beck et al., 2019). Relative to former precipitation products such as TRMM, MSWEP has better temporal coverage (starting in 1979), higher horizontal resolution (0.1°) and a similarly high temporal (3 hourly) resolution. Due to these advantages, MSWEP has recently been used as a primary source of information for assessments of mean and extreme precipitation (Alijanian et al., 2019; Lakew et al., 2020). In fact, several studies have already analyzed TCP using MSWEP (Torres-Alavez et al., 2021; Xiang et al., 2022).

The estimation of TCP requires a relatively costly computation. Firstly, a track, most frequently obtained from the International Best Track Archive for Climate Stewardship (IBTrACS), is used to transform the coordinate system of the gridded precipitation dataset into a cylindrical storm-relative system with the coordinates being the radius from storm-centre and their angle relative to the north or to the vertical wind shear vector. After this coordinate transformation, various metrics can be computed such as area-weighted averages of precipitation, calculations of the radius of maximum rain-rates (RMR), the az-

imuthally averaged structure of precipitation and estimates of the rainfall area (RA; see e.g. Prat and Nelson, 2016; Kim et al., 2018b; Vannière et al., 2020; Lavender and McBride, 2021; Tu et al., 2021).

These metrics are calculated for each track point, i.e., each TC observed every 3 or 6 hours. Therefore, processing TCP on high resolution datasets for the full coverage of the dataset (>40 years) has a relatively high computational cost, particularly in the newer and higher resolution datasets such as MSWEP. It is likely that many studies will repeat the same basic methodology

to diagnose TCP in MSWEP, as with TRMM, with some slight differences. These slight methodological differences may also make some results not directly comparable.

In this paper, we describe a newly developed dataset that merges the information of MSWEP with IBTrACS with the aim of making the dataset publicly available using standard reproducible methodology. This paper aims to save user-time by providing a freely-available dataset of various commonly used metrics of TCP, that is found in a user-friendly format and considerably

less extensive than the full MSWEP dataset. We therefore introduce the Global Multi-Source Tropical Cyclone Precipitation (MSTCP) dataset. The dataset contains two main elements in the form of tables: the main and profile datasets. The main dataset provides various TCP statistics per storm track for all basins of IBTrACS. The profile dataset comprises the azimuthal average precipitation by 10-km steps away from the eye of each storm (until 500 km), globally from IBTrACS.

The paper is structured as follows. Section 2 describes IBTrACS and MSWEP. Section 3 details the methodology and

implementation of the code. Section 4 describes the output (various precipitation metrics) of the newly introduced MSTCP dataset. Finally Section 5 analyzes the dataset per basin and intensity while Section 6 summarizes key results.

## 2 Data

We describe in this section the input data to build the MSTCP dataset, that is, IBTrACS and MSWEP.

### 2.1 IBTrACS

Tropical cyclone track information data is based upon the International Best Track Archive for Climate Stewardship (IBTrACS) version 4 (Knapp et al., 2010, 2018). We included all tracks from January 1979 until February 2023 from six cyclogenesis basins, that is, Atlantic (NA), Eastern North Pacific (EP), North Indian Ocean (NI), South Indian Ocean (SI), South Pacific (SP), Western Pacific (WP).

All the entries in IBTrACS are used for the TCP dataset regardless of the storm intensity, nature, or other status. The purpose

of this choice is to allow any potential user to subset the dataset using their own methodological choices for each unique purpose.

### 2.2 MSWEP

The Multi-Source Weighted-Ensemble Precipitation (MSWEP) version 2 dataset is a global precipitation product available at 3-hourly 0.1° from 1979 and onward with a global coverage over land and sea (Beck et al., 2019). MSWEP v2 is the only

global product at this resolution that is available on a 3-hourly latency for a period longer than 25 years. MSWEP is produced



through several steps that combine, calibrate and assimilate data from rain gauges, satellites and reanalyses. In total, MSWEP merges data from over 15 datasets including rain gauge, satellite and reanalysis data; see Beck et al. (2019) for further details on the precipitation datasets used by MSWEP and the merging algorithm.

One novel aspect of MSWEP is the inclusion of the period prior to the TRMM era, extending the record back to 1979. This
extension was achieved by including gauge, reanalysis and infrared derived estimates of precipitation. Another relevant quality of MSWEP is the way rain gauge and reanalysis data are calibrated correcting for reporting times mismatches in rain gauge data and precipitation frequency and cumulative distribution function (CDF) biases in reanalyses. The assessment of MSWEP by Beck et al. (2019) shows that this exhibits the best overall performance of several characteristics of precipitation over the continental United States and shows a plausible representation of the spatial pattern in mean, magnitude and frequency of
precipitation.

While the mean climatology of precipitation in MSWEP is more realistic than most other state-of-the-art satellite-derived products, a relatively good performance of the dataset in the mean-state climate might not be equally satisfactory for extreme precipitation. TCP rates typically extend into the extreme end of the distribution (>95th percentile) of observed precipitation. One key example is the study by Breña-Naranjo et al. (2015) that found that TRMM TMPA underestimated precipitation rates
compared to rain gauge data in Mexico. Moreover, the assessment of Jones et al. (2021) shows that disagreement between reanalyses datasets is higher for TCP than total precipitation. This means that there are relevant caveats about the fidelity of MSWEP v2 in representing TCP that must be considered when using this dataset.

## 3   Method

The core of the methodology lies on the intersection of each entry in the IBTrACS dataset (LAT, LON) with the corresponding
time (ISO_TIME) and precipitation field in MSWEP. Intersections are computed from January 1, 1979 to February 14, 2023. All scripts were coded in Python and Bash (see Section 7: Code and data availability) and the list of dependencies can be found in the files included in the `envs` folder.

The first analysis computes storm wide statistics using a 500 km radius geodesic buffer for each trajectory point. These metrics are the average TCP, maximum TCP rates, radius of maximum rain (RMR) and rainfall area. These values are stored in
the *main* dataset. Figure 1 illustrates the main metrics of the dataset. The full black circle in the middle represents the estimated center of hurricane Harvey on August 24, 2017 at 18:00 (UTC). The outermost circle represents the 500 km radius, a commonly used threshold to define TCP (Vannière et al., 2020; Zhang et al., 2021). This radius defines the circle where all statistics are computed. Precipitation outside this circle is ignored. The area-averaged TCP is therefore computed as the cumulative rain within that 500 km circle divided by the area of the circle (Lavender and McBride, 2021; García-Franco et al., 2023).

The fixed 500-km threshold as a radii to diagnose TCP has been extensively used in the literature (Jiang and Zipser, 2010; Villarini et al., 2014; Khouakhi et al., 2017; García-Franco et al., 2023, e.g.). Lavender and McBride (2021) shows that the 500-km threshold is, on average, the region of influence of TC on rainfall which roughly matches the scale of a TC cloud shield (Prat and Nelson, 2013). However, other studies have analyzed TCP using a flexible threshold that is case-by-case dependent.

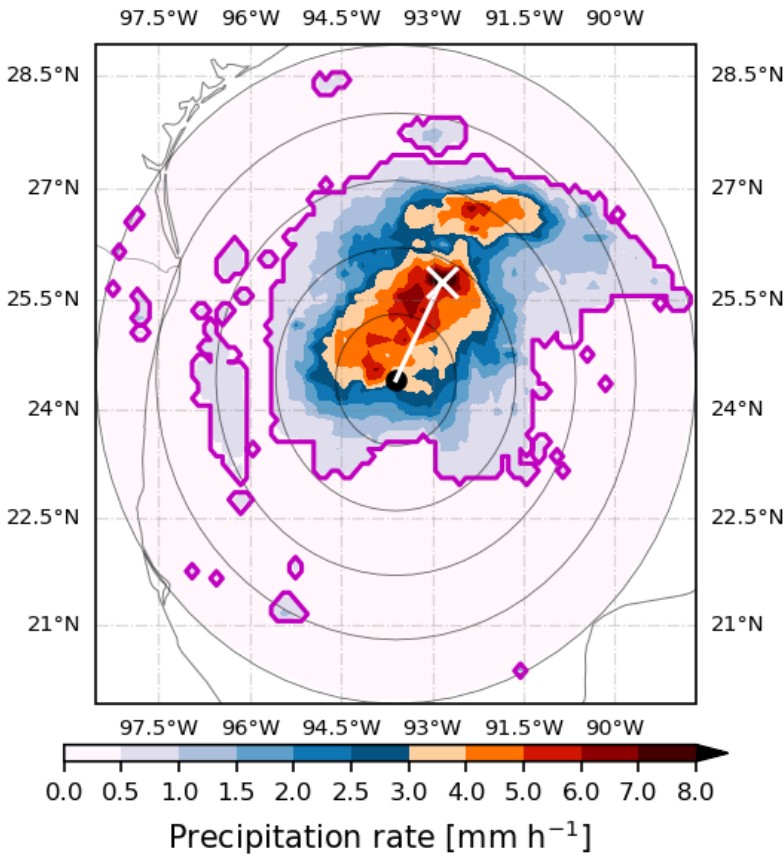

**Figure 1.** Illustration of the TCP metrics computed in the Global MSTCP dataset for Hurricane Harvey observed on August 24, 2017 at 18:00 (UTC). In the main dataset, the 500 km annuli (annotated as the outermost gray circle) is used to compute the mean precipitation. The white cross (X) denotes the location of maximum precipitation rates. The distance between the center and the white cross determines the radius of maximum rain. The binning of the data into radii increasing every 10 km is illustrated by annotating the 100, 200, 300 and 400 km radii. The rainfall area is shown as the magenta lines, as contours of the cells with at least 0.5 mm h$^{-1}$.





For example, Skok et al. (2013) used an object-based tracker and found that more detailed determination of TCP can lead to higher average precipitation rate estimates compared to fixed-threshold estimates. Our analysis uses the fixed 500-km threshold since it is arguably the more commonly used approach.

The rain area (RA) is defined as the area (km$^2$) within the circle where precipitation is above the threshold of 0.5 mm h$^{-1}$, following previous studies (Lin et al., 2015; Kim et al., 2018a; Zhao et al., 2018). This is illustrated in Figure 1 as the magenta contour. Other studies have chosen smaller (Guzman and Jiang, 2021) or larger thresholds (Yu et al., 2017; Niu et al., 2022) so that our choice is a relatively modest threshold for the RA.

The maximum precipitation rates, and their storm-relative location, for each snapshot derived from MSWEP is another key metric. In Figure 1, the white X represents the location of the maximum precipitation rates (within the 500 km circle), as in Lonfat et al. (2004). The line between X and the storm centre is therefore the radius of maximum rain (Lonfat et al., 2004; Guzman and Jiang, 2021).

In the second analysis, we compute azimuthally averaged statistics of TCP. This methodology aims to average out the variance of the precipitation that is unrelated to the radial coordinate and only considers the distance from storm center as the coordinate. For this purpose, we divided the 500 km radius geodesic buffer into 50 discrete non-overlapping geodesic ring buffers with a 10 km thickness to perform this analysis. Figure 1 illustrates this procedure by plotting equally distant (100 km) annuli from the storm, so that the calculation computes the average of all grid-points within each annulus. All raster cells whose centroids fall within the geodesic buffer are kept to compute the azimuthally averaged TCP rates (Kim et al., 2018b; Lavender and McBride, 2021; Moon et al., 2022). These metrics are stored in the *profile* dataset.

To mitigate the unequal grid cell area problem, statistics were area-weighted. Grid cell areas in km-sq were estimated using the *Climate Data Operators* `gridarea` operator. (Schulzweida, 2022). Distance conversion from degrees to kilometers were computed with the Haversine formula with an Earth radius of 6371 km.

To analyse the original MSWEP dataset, compute statistics, assemble and save both CSV files, the computation time required is approximately 200 hours on a 12-core/24-thread (AMD Ryzen 9 3900x) computer with 256 GB of RAM (4 x 32 GB DDR4) with a 4TB HDD (5400RPM SATA 6Gb/s 64MB Cache (RoHS)) under Ubuntu 20.04.6 LTS.

## 4 The Global MSTCP dataset

The Global MSTCP dataset (Morin et al., 2023) is made of two CSV files: *main* and *profile* datasets. The columns for both files are detailed below. The *main* dataset comprises precipitation statistics presented in a format similar to IBTrACS, where each row corresponds to data per 3-hour segment. The *main* CSV file size is about 40MB. The *profile* dataset comprises azimuthally averaged precipitation statistics in 10 km bins from storm center out to 500 km. Each row corresponds a 3-hour segment with a given distance from the eye (10, 20, ..., 500) km. The *profile* CSV file size is about 1.3GB. The dataset is freely available for download at https://zenodo.org/records/10105751 and can also be found with the doi 10.5281/zenodo.10105751.





## 4.1 Main dataset

The columns of the *main* dataset are:

- `ISO_TIME`: from IBTrACS, ISO time in Universal Time Coordinates (UTC) of the corresponding TC location (latitude, longitude);

- `LAT`: from IBTrACS, latitude (degrees) of the eye of the storm (LAT in IBTrACS, which is the average latitude reported by several stations);

- `LON`: from IBTrACS, longitude (degrees) of the eye of the storm (LON in IBTrACS, which is the average longitude reported by several stations);

- `NAME`: from IBTrACS, name of the storm;

- `SID`: from IBTrACS, unique storm identifier;

- `row_id`: unique row identifier to link the *main* and *profile* datasets;

- `variant`: from MSWEP, one of the variants available: "Past" or "NRT" (see the July 2023 Expert Developer Guidance section from the NCAR Climate Data Guide, (Beck et al., 2023))

- `rain_area`: rain area (km$^2$) over a threshold of 0.5 mm/hr.

- `area_avg_TCP`: cumulative rain in a circle of 500 km from the eye averaged over rain area (mm/h / km$^2$).

- `lat_max_precip`: latitude (degrees) of the location of the maximum precipitation;

- `lon_max_precip`: longitude (degrees) of the location of the maximum precipitation;

- `max_precip`: maximum precipitation (mm/hr) found within the circle of 500 km from the eye.

- `RMR`: distance (km) between the eye of the storm and the location of the maximum precipitation found (radius of maximum rain).

## 4.2 Radial profile dataset

The radial profile dataset provides estimates of the azimuthally averaged precipitation which are often described as radial profiles of precipitation. In addition to `ISO_TIME`, `LAT`, `LON`, `SID`, `row_id` and `variant`, columns for the *radial profile* dataset include:

- `bin`: distance $r$ (km) from the eye of the storm. A value of $r$ indicates the statistic is computed between $r$ to $r + 10$ km from the eye of the storm, with $r = 0, 10, 20, \ldots, 490$.

- `azim_avg_TCP`: azimuthal average precipitation computed between $r$ to $r + 10$ km from the eye of the storm.





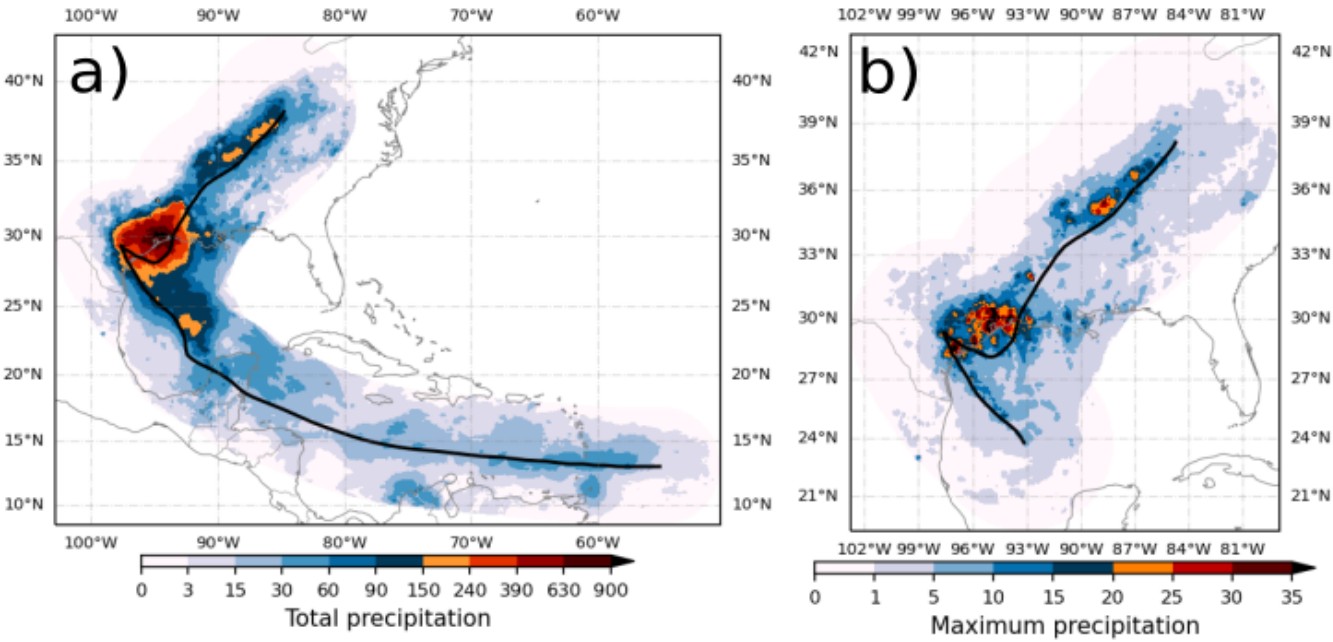

**Figure 2.** Hurricane Harvey (2017) estimated TCP in MSWEP. (a) Accumulated precipitation amounts [mm] and track (solid line). (b) Maximum hourly precipitation rates [mm h$^{-1}$].

## 5    Applications

This section provides key descriptive statistics from the *main* and *profile* datasets, in addition to a case study of Hurricane Harvey.

### 185    5.1    Hurricane Harvey

Hurricane Harvey flooded Houston and nearby areas in 2017 due to enormous rainfall amounts and precipitation rates. The floods associated with Hurricane Harvey were made more likely by greenhouse warming that increased the total amount of water vapour available in the atmosphere (Emanuel, 2017) and more damaging by urbanization (Zhang et al., 2018). Figure 2a shows the total amount of precipitation associated with Hurricane Harvey as estimated by our analysis of MSWEP. Most of the precipitation was observed during landfall in Texas, where the TC turned southeast for a short time followed by a northward trajectory into mainland United States. The stalling of Harvey around the area was devastating due to the very high rainfall rates (Fig. 2b) and the relatively long time Harvey was around the Houston area.

The spatial distribution and rainfall total amounts estimated by MSWEP agree well with those reported in previous studies using rain-gauge, radar and other satellite-derived products (Van Oldenborgh et al., 2017). The spatial distribution of the maximum precipitation rates (Fig. 2b) also agree with the results of Van Oldenborgh et al. (2017). This demonstrates that our estimates of average TCP and maximum TCP are consistent with previous literature.

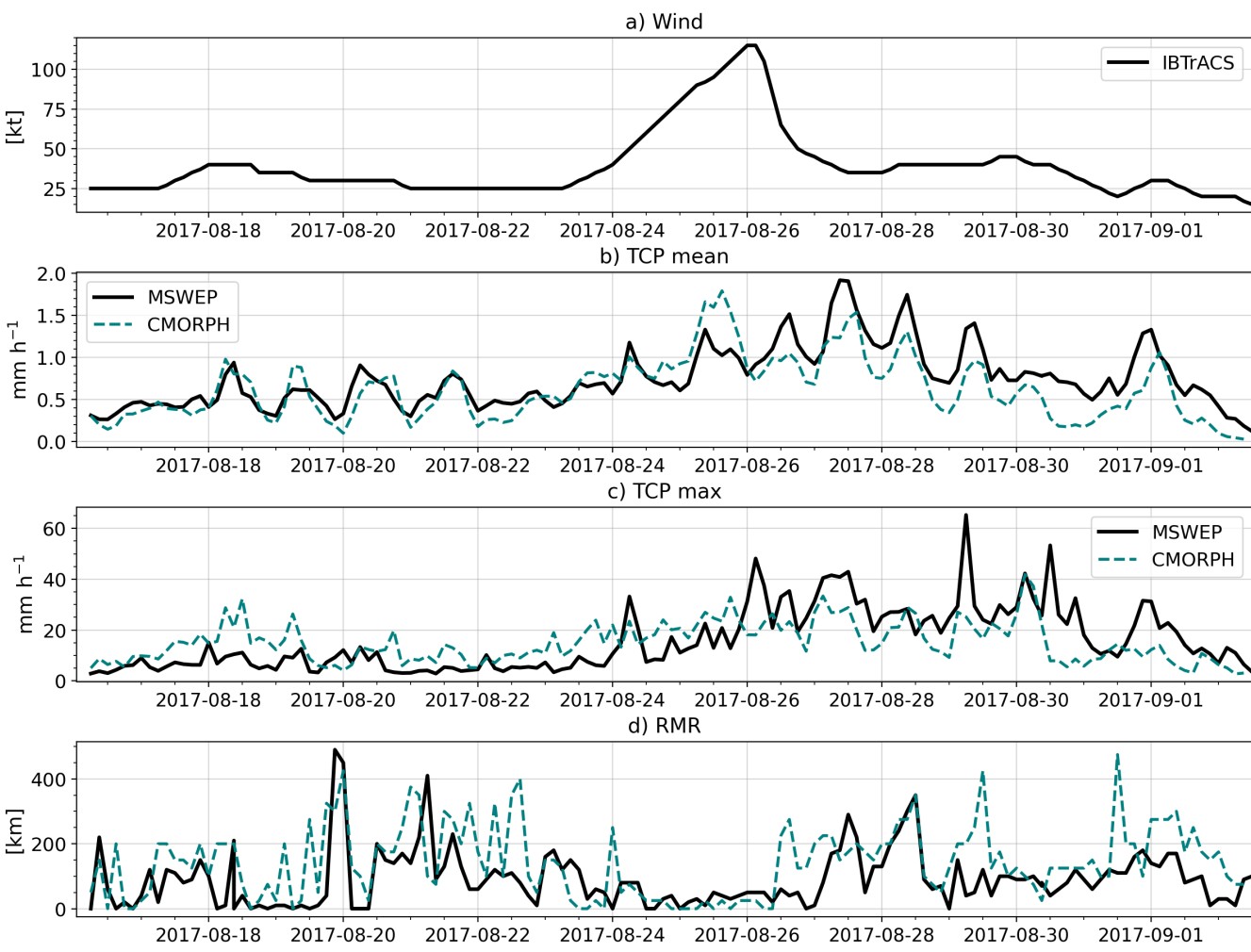

**Figure 3.** Time series of (a) maximum sustained wind speed [kt], (b) mean TCP, (c) maximum TCP [mm h$^{-1}$] and (d) RMR [km] in Hurricane Harvey (2017) estimated from (a) IBTrACS and (b-d) CMORPH and MSWEP.





**Table 1.** Mean statistics separated per TC basin. Shown are total entries (counts) in the dataset, and the basin average of mean TCP (in mm h$^{-1}$), max TCP (in mm h$^{-1}$) and the RMR (in km).

| BASIN | Count | TCP$_{mean}$ [mm h$^{-1}$] | TCP$_{max}$ [mm h$^{-1}$] | RMR [km] | RA [$10^5$ km$^2$] |
|---|---|---|---|---|---|
| NA | 38277 | 0.78 | 12.8 | 94 | 2.72 |
| EP | 45309 | 0.68 | 11.3 | 92 | 2.42 |
| NI | 12844 | 0.93 | 14.3 | 95 | 3.28 |
| SI | 39249 | 0.97 | 12.8 | 113 | 3.33 |
| SP | 20896 | 1.18 | 14.3 | 117 | 3.64 |
| WP | 100364 | 1.18 | 13.8 | 100 | 3.94 |

To compare the results obtained from MSWEP, we have repeated the analysis of Hurricane Harvey using the CMORPH dataset (Joyce et al., 2004; Xie et al., 2019). CMORPH has recently been used for several model assessments of TCP (Moon et al., 2022; García-Franco et al., 2023) and is a dataset of relatively high resolution (0.25°).

In Figure 3 the time series of the main metrics diagnosed with MSWEP for Hurricane Harvey (2017) agree well with CMORPH. The estimates of the average precipitation within 500 km of the center of the storm are very similar between the two datasets. The maximum precipitation rate estimates, however, are slightly higher in MSWEP, especially during the period of highest intensity of Harvey. This may be due to the higher spatial resolution of MSWEP (0.1°) compared to CMORPH (0.25°). Even though it appears that both datasets agree less in their estimates of RMR during the period of maximum intensity,

both datasets suggest a very low RMR. Overall, our analysis shows that the estimates of TCP for Hurricane Harvey from MSWEP agree well with those of CMORPH.

### 5.2 Descriptive statistics of the main dataset

This section reports statistics of the main dataset and derived metrics. The four metrics that compose the main dataset are the mean TCP, maximum TCP rates, RMR and the rainfall area (RA). Table 1 reports the number of entries found in each

TC basin across the dataset, as well as the average values of the chosen TCP metrics. The number of counts per basin shows that the dataset includes a considerable amount of cases which may prove useful to define climatologies conditioned on storm characteristics, basin or environmental conditions. Differences in the counts of TCs between basins in the dataset is due to the different number of observed TCs in each basin, as more TCs are observed in the WP and less TCs in the NI for example.

Various studies have shown how these different TCP metrics vary spatially, due to environmental conditions (Lavender and

McBride, 2021), and also due to storm scale processes (Kim et al., 2018b; Moon et al., 2022), that usually are associated with mean storm intensity. However, the TCP versus storm intensity relationship is highly non-linear in observations (Touma et al., 2019). A preliminary assessment of the inter-basin differences in these metrics is given in Table 1 and Figure 4.

Table 1 shows how each metric varies with TC basin. The average TCP in a 500 km radius is highest in the SP and WP and lowest in the EP on average which may be largely due to median storm intensity differences (Lavender and McBride,

2021). The maximum TCP rates are highest in the NI and SP and lowest in the EP, which based on the analysis of Lavender

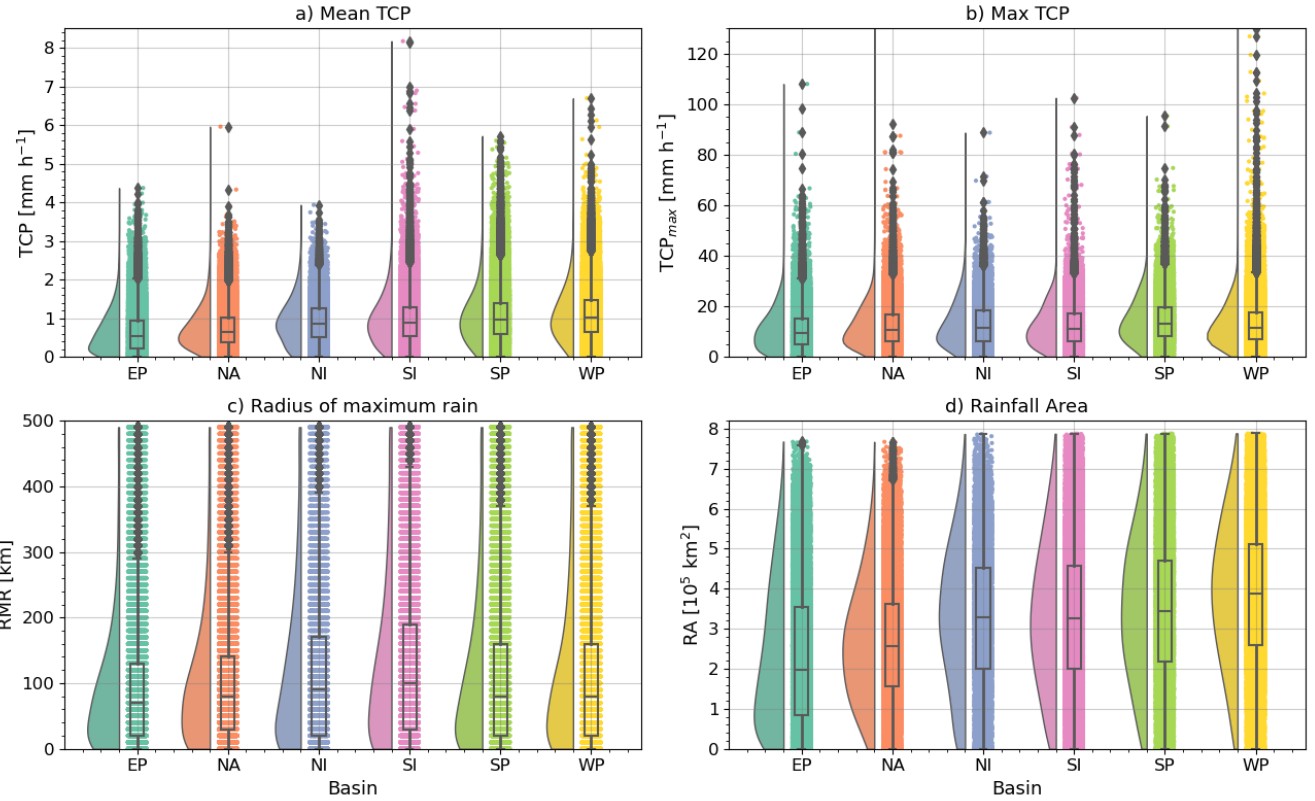

**Figure 4.** Rain cloud plot showing the violin and box plots of the distribution of (a) mean TCP in 500 km radii in mm h$^{-1}$, (b) maximum TCP mm h$^{-1}$, (c) radius of maximum rain and (d) rainfall area. Boxplots show the interquartile range [box], the spread [whiskers], and outliers are shown as diamonds.

and McBride (2021) is due to the differences in the eyewall precipitation rates between these basins. The high NI maximum precipitation rates is nonetheless surprising since the median TC intensity is relatively lower in the NI basin compared to other basins. The RA is highest in the WP, indicative of a larger region of influence of the TC, and lowest in the EP and NA, which also agrees well with the results of Lavender and McBride (2021) for inter-basin differences in TC size.

Figure 4 illustrates the statistical distribution of these 4 metrics conditioned on the TC basin. While the median of the area-averaged TCP values is highest in the SP and WP, the outliers in the SI are highest among all basins. The WP TCs show the highest third quartile and outliers of TCP$_{max}$ although the mean value of this metric is highest in the NI. The interpretation of this result highlights the non-gaussian nature of the TCP distribution and emphasizes the need to understand extreme cases of very high TCP.

The RMR inter-basin differences (Fig. 4c) are the result of basin-wide differences in intensity which modulates the radius of maximum wind (RMW) as stronger storms tend to have a stronger eyewall closer to the storm centre but also in TC size which modulates the extent of the TC cloud shield and RA. Based on the RA, TCs have the smallest size in the EP as the majority of


**Table 2.** Mean statistics separated per TC intensity category. As in Table 1 but for different storm intensity categories based on the Saffir-Simpson wind scale.

| Cat. | Percent | $TCP_{mean}$ [mm h$^{-1}$] | $TCP_{max}$ [mm h$^{-1}$] | RMR [km] | RA [10$^4$ km$^2$] |
|------|---------|------------------|------------------|----------|----------------|
| TD | 30.2 | 0.81 | 10.9 | 127 | 31.1 |
| TS | 35.6 | 0.98 | 13.5 | 91.4 | 32.8 |
| Cat 1 | 11.3 | 1.17 | 15.3 | 67.5 | 35.7 |
| Cat 2 | 4.26 | 1.27 | 16.6 | 66.6 | 37.1 |
| Cat 3 | 5.0 | 1.31 | 16.9 | 63.6 | 38.5 |
| Cat 4 | 3.5 | 1.5 | 17.7 | 60.6 | 43.4 |
| Cat 5 | 0.73 | 1.73 | 18.2 | 59.3 | 50.7 |

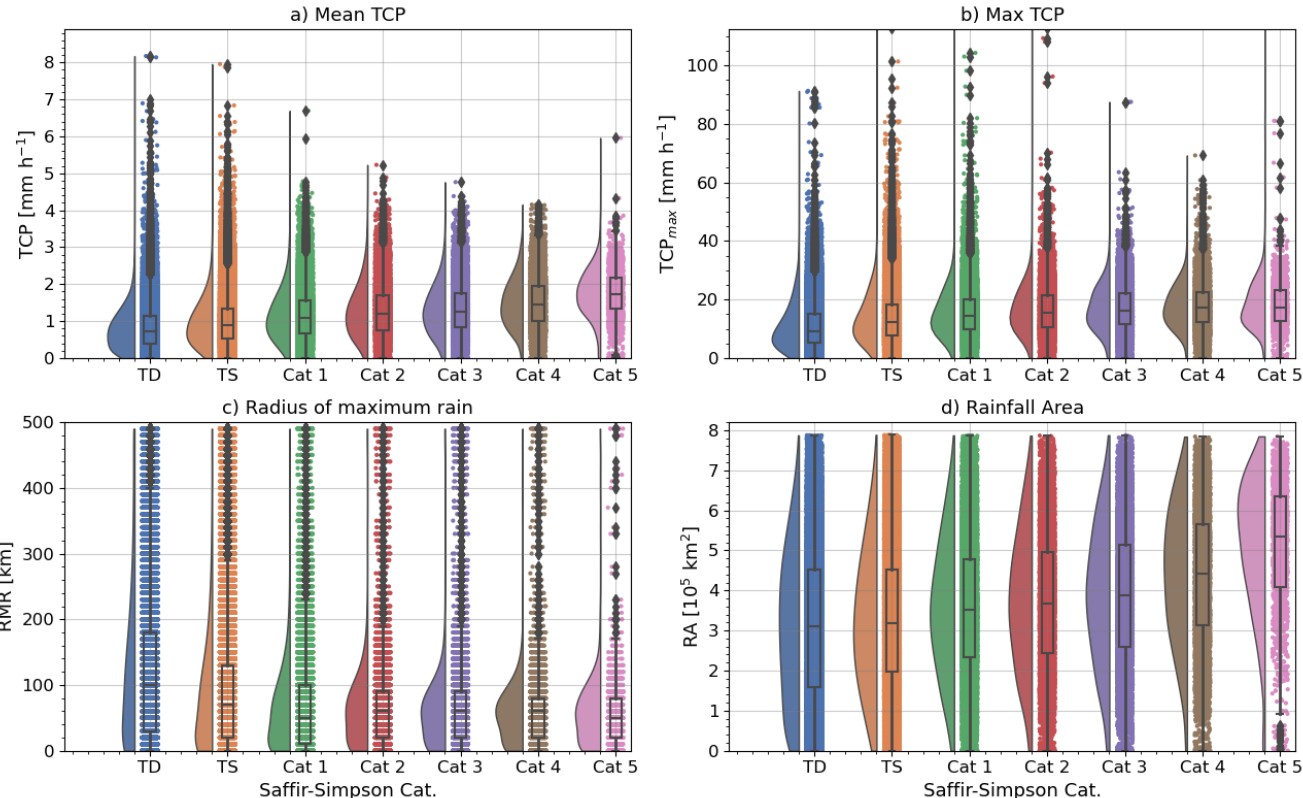

**Figure 5.** As in Figure 4 but distributions are conditioned on the storm intensity categories per the Saffir-Simpson wind scale.

cases are consistently smaller than in basins such as WP and SP. These results are indicative of differences in the average storm size, water vapour content and intensity between basins (Skok et al., 2013; Touma et al., 2019; Lavender and McBride, 2021).





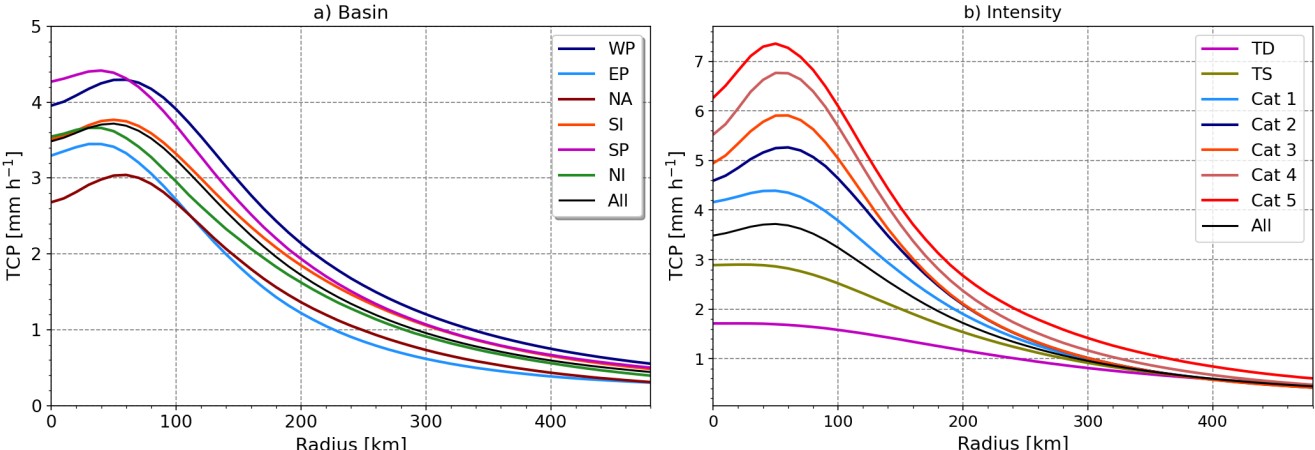

**Figure 6.** Azimuthally averaged precipitation as a function of distance from storm centre [Radius in km] for cases separated by (a) basin and (b) intensity category. In b) only cases with a maximum sustained wind speed of 34 kt are considered.

The relationship between TCP and storm intensity is summarized in Table 2 and illustrated in Figure 5. Even though storm intensity does not explain all of the variability in TCP, the average and maximum TCP rates, as well as RA, increase with intensity whereas the RMR decreases with intensity. A smaller RMR is consistent with a tighter radius of maximum wind (RMW) which is also observed in stronger storms (Didlake and Houze, 2013). The maximum TCP rates are also larger for Cat. 4-5 storms compared with Cat. 1 and 2, although Figure 5 shows that Cat. 1 through 5 TCs can have precipitation rates exceeding $50 \, \mathrm{mm \, h^{-1}}$.

Similarly, stronger storms have a smaller RMR (Fig. 5c), consistent with a stronger primary circulation. Stronger storms also show a larger RA (Fig. 5d), consistent with a larger secondary circulation. Even though weak storms have an average area smaller than stronger storms, TS and Cats. 1 and 2 storms frequently show very large RAs. Weak storms with large RA also emphasize that rain-related TC impacts from large RA may not be solely driven by storm intensity.

The differences in the distributions depicted in Figure 5 show that while, on average, these TCP metrics are related to the storm intensity, TCP variability is modulated by other factors. Relatively weak storms can still cause significant floods and exhibit very large precipitation rates. These results, however, demonstrate the potential use of this dataset in the investigation of the environmental or storm scale factors that modulate these metrics.

### 5.3 Statistics of the profile dataset

In addition to the main dataset, the profile dataset provides the azimuthally averaged precipitation in 10 km bins from 0 to 500 km. This means that for each entry in the main dataset, there are 50 entries in the profile dataset where each row is the averaged precipitation in a given radial bin for a specific TC case.





The radial structure of precipitation is useful to diagnose the storm structure in observed or simulated TCs (see e.g. Kim et al., 2018b; Moon et al., 2022; García-Franco et al., 2023). Figure 6 shows the azimuthally averaged structure of TCP in all

cases separated by basin and intensity category. One key feature of the radial profiles of TCP computed from MSWEP is that the inner core structure of precipitation is better resolved than in previous datasets. The radial profile of TCP is characterized by maximum precipitation rates around 50-100 km, where the RMW is typically located, and TCP decreases both into the storm centre, where the eye is usually drier, and away from the storm centre where ascent is weaker and precipitation linked to the rain bands (Didlake and Houze, 2013). In MSWEP, with a 0.1° resolution, this structure is diagnosed in storms of at least

TS status. This structure could not be diagnosed in previous coarser observational datasets.

Figure 6 shows that the averaged rainfall structure varies from basin to basin but the dependence on storm intensity is stronger. We observe that TCs in the WP have the highest TCP rates in the inner core. While TCs in the EP have higher TCP in the inner core, the NA storms have higher precipitation rates at larger radii. These results illustrate the potential use of this profile dataset to further diagnose, from a process-level understanding, observed differences in the total precipitation associated

with the azimuthally averaged structure of a TC.

## 6 Summary

The Global MSTCP dataset provides estimates of TCP as measured by MSWEP v2 following the TC tracks provided in the IBTrACS dataset. Key variables that are found in the main dataset are average TCP within 500 km of storm centre, the maximum precipitation rates, the RMR and the area of rainfall covered by the storm. In addition, a profile dataset includes

the azimuthally averaged TCP, which is often used to diagnose the storm scale structure of precipitation within a TC. The frequency of the MSWEP dataset is 3-hourly with a global coverage from 1979 and onward which is a significant increase in the coverage period from previous datasets used for TCP studies.

Examples of the output of the dataset show that our treatment of MSWEP is able to replicate the spatial pattern and rainfall amounts associated with Hurricane Harvey that have been reported by previous studies. The time series of average TCP in

Harvey diagnosed from MSWEP agrees well with results from CMORPH. This is evidence that MSWEP will be useful to diagnose TCP in future studies.

The mean statistics of the dataset separated by TC basin and storm intensity agree well with our knowledge of the differences in the characteristics of TCP per basin and the TCP versus storm intensity relationships presented in previous studies using other precipitation products. This demonstrates the potential use of our dataset for future studies including model evaluation,

process-based understanding, diagnosing trends, or investigating case-studies of TCP.

Given that our dataset compresses the spatial dimension into zero dimensional quantities, except for the azimuthally averaged structure of precipitation, the dataset is not meant to be used to reconstruct the global spatial distribution of TCP and may limit its use in some studies that require detail on the spatial distribution of TCP. Our analysis of the maximum precipitation rates in Hurricane Harvey shows how two satellite-derived datasets render similar estimates of average TCP but agree less on extreme

precipitation rates. This also highlights the uncertainty of using satellite-derived datasets to estimate extreme precipitation



rates. This dataset will be useful for the tropical cyclone research and wider scientific community by making it straightforward for any user to have a freely available, that is easily portable, global dataset of TCP metrics from 1979 and onward.

## 7 Code and data availability

The Python code to build the main and profile datasets are provided here: https://github.com/GabrielMorin1109/MSTCP-Dataset.

The main and profile datasets (Morin et al., 2023) can be downloaded from Zenodo here: https://doi.org/10.5281/zenodo.10105751 with the following doi doi:10.5281/zenodo.10105751.

Other datasets are also freely available such as IBTrACS (Knapp et al., 2018) and CMORPH (Xie et al., 2019).

*Author contributions.* Conceptualization : MB;

Data curation : GM, JLGF;

Formal analysis : GM, JLGF;

Funding acquisition : MB;

Investigation : GM, JLGF;

Methodology : GM, MB, JLGF;

Project administration : MB;

Resources : MB;

Software : GM, JLGF;

Supervision : MB;

Validation : GM, JLGF;

Visualization : GM, JLGF;

Writing – original draft preparation : MB, JLGF;

Writing – review & editing : GM, MB, JLGF;

*Competing interests.* Morin, Boudreault and García-Franco each declare they have no competing interests.

*Acknowledgements.* Boudreault acknowledges funding from the Natural Sciences and Engineering Research Council of Canada (NSERC)

through the Discovery Grant program (RGPIN 2021-03362). Gabriel Morin acknowledges financial support from the NSERC Undergraduate Student Research Award program. García-Franco acknowledges support from the NASA MAP program (80NSSC21K1495).

The authors would like to thank Francesco S.R. Pausata (UQAM) and Christian Dominguez (UNAM) for comments on an earlier version of this manuscript.





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
