# Peer review of "A Global Multi-Source Tropical Cyclone Precipitation (MSTCP) Dataset"

_Earth System Science Data, 2023_

## Author Comment (AC1)

**ESSD-2023-460 A Global Multi-Source Tropical Cyclone Precipitation (MSTCP) Dataset**

**Author's answers to each referee's comments**

**Referee # 1**

**We thank the referee for taking the time to review the manuscript.**

The authors produce a so-called Global Multi-Source Tropical Cyclone Precipitation (MSTCP) dataset by using two publicly available datasets: a tropical cyclone (TC) best track dataset (IBTrACS) and a new version of global precipitation product (MSWEP V2), and then performed some statistics of TC precipitation, most of which have already been investigated by using similar datasets in previous studies.

**Nothing to say.**

Although the computation is relatively cost, the MSTCP dataset can be easily derived by professionals and thus it is not unique.

**As the reviewer notes, the calculations required to analyze TCP are costly and time-consuming. If each user repeats our analysis, the community will spend an overall large amount of time calculating these diagnostics. Instead, our dataset allows a larger community of users to devote their time on the direct analysis of TCP and avoid the very high initial cost of calculating TCP diagnostics. By making the MSTCP dataset available, we aim to accelerate knowledge transfer and creation. Given that TRMM has been discontinued in 2019, it is likely that MSTCP (based upon MSWEP v2) will become the standard in the future for TCP analysis.**

**In addition, the MSTCP dataset is unique in the sense that it is the first publicly available dataset of processed TCP from a global precipitation dataset. We found that Zhang et al. (2019) and Torres-Alavez et al. (2021) did apply MSWEP v2 and IBTrACS to process TCP but they both did not publish the resulting dataset. Moreover, Torres-Alavez et al. (2021) only covers 1995-2014, whereas MSTCP is available from 1979 to the present. A longer time coverage is more useful, e.g., for trend analyses. Finally, Zhang et al. (2019) has a longer time coverage (1980-2015) but they only computed precipitation over land.**

In addition, the azimuthally averaged MSTCP data limit its usefulness in studying TC precipitation asymmetry.

**Azimuthal averages are commonly used for model evaluation (see e.g. Vannière et al., 2020, Moon et al., 2022) and observational studies interested in the spatial or temporal variability of the observed structure of azimuthally averaged TCP, including the study of trends of TCP (see e.g., Lavender and**

**McBride 2021, Tu et al., 2021). These applications provide a strong motivation for our dataset to provide an azimuthally averaged product.**

**In addition, we could add additional columns in the profile dataset to report the radial profiles of precipitation per quadrant (shear-relative and storm-relative) to expand the usefulness of the MSTCP dataset.**

Therefore, I don't think the manuscript qualifies publication in ESSD.

**Nothing to say.**

**References**

- Lavender, Sally L., and John L. McBride. "Global climatology of rainfall rates and lifetime accumulated rainfall in tropical cyclones: Influence of cyclone basin, cyclone intensity and cyclone size." *International Journal of Climatology* 41 (2021): E1217-E1235.
- Moon, Yumin, Daehyun Kim, Allison A. Wing, Suzana J. Camargo, Ming Zhao, L. Ruby Leung, Malcolm J. Roberts, Dong-Hyun Cha, and Jihong Moon. "An evaluation of tropical cyclone rainfall structures in the HighResMIP simulations against satellite observations." *Journal of Climate* 35, no. 22 (2022): 7315-7338.
- Torres-Alavez, J. A., Glazer, R., Giorgi, F., Coppola, E., Gao, X., Hodges, K. I., Das, S., Ashfaq, M., Reale, M., and Sines, T.: Future projections in tropical cyclone activity over multiple CORDEX domains from RegCM4 CORDEX-CORE simulations, Climate Dynamics, 57, 1507–1531, 2021.
- Tu, Shifei, Jianjun Xu, Johnny CL Chan, Kian Huang, Feng Xu, and Long S. Chiu. "Recent global decrease in the inner-core rain rate of tropical cyclones." *Nature communications* 12, no. 1 (2021): 1948.
- Vannière, Benoît, Malcolm Roberts, Pier Luigi Vidale, Kevin Hodges, Marie-Estelle Demory, Louis-Philippe Caron, Enrico Scoccimarro, Laurent Terray, and Retish Senan. "The moisture budget of tropical cyclones in HighResMIP models: large-scale environmental balance and sensitivity to horizontal resolution." *Journal of Climate* 33, no. 19 (2020): 8457-8474.
- Zhang, W., Villarini, G., Vecchi, G. A., & Murakami, H. (2019). Rainfall from tropical cyclones: high-resolution simulations and seasonal forecasts. *Climate dynamics*, *52*, 5269-5289.

---

## Author Comment (AC2)

**ESSD-2023-460 A Global Multi-Source Tropical Cyclone Precipitation (MSTCP) Dataset**

**Referee # 2**

**We thank the referee for having taken the time to review the manuscript.**

The topic of this manuscript is greatly interesting. A good dataset of tropical Cyclone Precipitation information is vital for accurately understanding the varying Cyclones under the changing climate. ESSD is a big journal for publishing high quality datasets.

**Nothing to say.**

However, after carefully reading this manuscript, I found there are various points confusing me, including but not limited to: (1) there is very limited descriptions on the scientific methodology, so I can not judge the reasonability;

**It would be useful if the referee provided specific examples where the methodology needs a more detailed description. We would be happy to add further details where needed.**

(2) the MSTCP data has not been validated in terms of its accuracy, so at what extent do I believe it;

MSTCP is based upon MSWEP v2 which is a precipitation product made from an ensemble of precipitation datasets. MSWEP v2 has been validated and used by several studies of precipitation in the tropics and subtropics (Prakash et al., 2019, Xu et al., 2019, Chua et al., 2022, Mekonenn et al., 2023). These studies conclude MSWEP v2 has good performance, and in most cases a better accuracy, when compared to other available products

**IBTrACS** is the sole dataset for global tropical cyclone tracks and is widely used in the TC community.

We made a comparison of MSTCP with CMORPH in Section 5.1 for hurricane Harvey with a very satisfying performance. It is true that CMORPH and MSWEP share some input products in their merging methodologies, which means that they are not fully independent datasets. MSWEP is based on the merger of an exhaustive list of input products, which means it is not straightforward to conduct a validation with a fully independent dataset. Section 5.2 also compares the TCP statistics with the literature, thus confirming its performance.

It would be similarly useful if the reviewer could specify what kind of validation and against which products they consider to be necessary. Our goal is to improve the dataset and manuscript as best we can. and (3) the writings are really not good enough for the big journal, ESSD, while the current manuscript is more like a document than a scientific paper.

It was written as a dataset paper, not as a scientific paper. We used the provided template for ESSD. The format is similar to previously published ESSD manuscripts (see e.g., Sun and Fu 2021, Han et al., 2023, Beguería et al., 2023), including Roca et al, 2019, which provides an illustration of the FROGs dataset and not an extensive validation of each of the components of the dataset. Moreover, our rationale of providing a processed dataset merging already available products, with a specific focus on precipitation, to accelerate data transfer and data processing fits well within the scope of recently published papers in the ESSD journal (Sun and Fu, 2021, Beguería et al., 2023).

Therefore, I am very sorry for not recommending it for publication at this stage.

**References**

- Beguería, Santiago, Dhais Peña-Angulo, Víctor Trullenque-Blanco, and Carlos González-Hidalgo. "MOPREDAScentury: a long-term monthly precipitation grid for the Spanish mainland." *Earth System Science Data* 15, no. 6 (2023): 2547-2575.
- Chua, Zhi-Weng, Alex Evans, Yuriy Kuleshov, Andrew Watkins, Suelynn Choy, and Chayn Sun. "Enhancing the Australian Gridded Climate Dataset rainfall analysis using satellite data." *Scientific Reports* 12, no. 1 (2022): 20691.
- Han, Jingya, Chiyuan Miao, Jiaojiao Gou, Haiyan Zheng, Qi Zhang, and Xiaoying Guo. "A new daily gridded precipitation dataset for the Chinese mainland based on gauge observations." *Earth System Science Data* 15, no. 7 (2023): 3147-3161.
- Mekonnen, Kirubel, Naga Manohar Velpuri, Mansoor Leh, Komlavi Akpoti, Afua Owusu, Primrose Tinonetsana, Tarek Hamouda, Benjamin Ghansah, Thilina Prabhath Paranamana, and Yolande Munzimi. "Accuracy of satellite and reanalysis rainfall estimates over Africa: A multi-scale assessment of eight products for continental applications." *Journal of Hydrology: Regional Studies* 49 (2023): 101514.
- Prakash, Satya. "Performance assessment of CHIRPS, MSWEP, SM2RAIN-CCI, and TMPA precipitation products across India." *Journal of hydrology* 571 (2019): 50-59.
- Roca, Rémy, Lisa V. Alexander, Gerald Potter, Margot Bador, Rômulo Jucá, Steefan Contractor, Michael G. Bosilovich, and Sophie Cloché. "FROGS: a daily 1× 1 gridded precipitation database of rain gauge, satellite and reanalysis products." *Earth System Science Data* 11, no. 3 (2019): 1017-1035.
- Sun, Lilu, and Yunfei Fu. "A new merged dataset for analyzing clouds, precipitation and atmospheric parameters based on ERA5 reanalysis data and the measurements of the Tropical Rainfall Measuring Mission (TRMM) precipitation radar and visible and infrared scanner." *Earth System Science Data* 13, no. 5 (2021): 2293-2306.
- Xu, Zhengguang, Zhiyong Wu, Hai He, Xiaotao Wu, Jianhong Zhou, Yuliang Zhang, and Xiao Guo. "Evaluating the accuracy of MSWEP V2. 1 and its performance for drought monitoring over mainland China." *Atmospheric research* 226 (2019): 17-31.